# Recent Advances in Nuclear Imaging of Receptor Expression to Guide Targeted Therapies in Breast Cancer

**DOI:** 10.3390/cancers11101614

**Published:** 2019-10-22

**Authors:** Barbara Salvatore, Maria Grazia Caprio, Billy Samuel Hill, Annachiara Sarnella, Giovanni Nicola Roviello, Antonella Zannetti

**Affiliations:** Istituto di Biostrutture e Bioimmagini-CNR, 80145 Naples, Italy; barbara.salvatore@cnr.it (B.S.); mariagrazia.caprio@ibb.cnr.it (M.G.C.); billy.hill@ibb.cnr.it (B.S.H.); achiara.sarnella@gmail.com (A.S.); giroviel@unina.it (G.N.R.)

**Keywords:** breast cancer, receptor status, molecular imaging

## Abstract

Breast cancer remains the most frequent cancer in women with different patterns of disease progression and response to treatments. The identification of specific biomarkers for different breast cancer subtypes has allowed the development of novel targeting agents for imaging and therapy. To date, patient management depends on immunohistochemistry analysis of receptor status on bioptic samples. This approach is too invasive, and in some cases, not entirely representative of the disease. Nuclear imaging using receptor tracers may provide whole-body information and detect any changes of receptor expression during disease progression. Therefore, imaging is useful to guide clinicians to select the best treatments for each patient and to evaluate early response thus reducing unnecessary therapies. In this review, we focused on the development of novel tracers that are ongoing in preclinical and/or clinical studies as promising tools to lead treatment decisions for breast cancer management.

## 1. Introduction

Breast cancer is a malignant disease with the highest incidence in women worldwide [1]. There is a wide variation in patient prognosis and outcomes due to the high heterogeneity of this carcinoma. The three main biomarkers of interest in this cancer include estrogen receptor (ER), progesterone receptor (PR), and human epidermal growth factor receptor 2 (HER2). Nowadays, at least four molecular subtypes have been identified with different levels of expression of these receptors: luminal A, luminal B, HER2-enriched, and basal-like. Triple-negative breast cancer (TNBC), which is a part of the basal-like subgroup, is characterized by the lack of PR, ER, and HER2 expression, and by extremely high mortality due to metastatic and drug-resistance recurrent disease [2]. The ER is the major driver of breast cancer being expressed in 75% of patients and tamoxifen that binds ER was the first drug approved for metastatic breast cancer [3]. HER2 is overexpressed in approximately 15% to 25% of patients at initial diagnosis and is related with cancer aggressiveness and poor survival. The development of specific molecules, which target this receptor such as trastuzumab and pertuzumab or the tyrosine kinase inhibitor lapatinib, has significantly improved the outcome in HER2-positive breast cancer patients [3]. Both tumor heterogeneity and the occurrence of resistance to treatments may fundamentally affect receptor status and therefore response to therapy. To overcome this issue, recent investigations are focusing on the characterization of other well-known receptors involved in breast cancer progression as potential therapeutic and imaging targets such as tyrosine kinases receptors, integrin receptors, chemokine receptors, immune checkpoint receptors, somatostatin receptors, gastrin releasing peptide receptors, and neuropeptide Y receptors. Over the past decades, many molecules specifically targeting them were developed: monoclonal antibodies, antibody fragments, nanobodies, small kinase inhibitors, peptides, and oligonucleotide aptamers. At present, several of these drugs are already in clinical trials while others are under investigation in pre-clinical models. 

Current practice guidelines use biopsy and immunohistochemistry to define the receptors status in cancer tissues and to guide subsequent treatments. The integration of molecular imaging into breast cancer daily management could improve the standard of care, providing additional information on the heterogeneity of tumor lesions and facilitating early diagnosis, accurate staging, and personalized treatment planning. In particular, it could improve the management of metastatic breast carcinoma where receptor status can change during disease course and repeat biopsies may be too invasive for the patients [4]. Functional nuclear imaging using single photon emission computed tomography (SPECT) and positron emission tomography (PET) could allow the visualization of molecular changes in receptor status in primary tumor, metastases, and throughout treatments [5,6]. The main purpose of this review is to gain an overview of the relevant findings on the development of novel radiotracers targeting receptor status in breast cancer that are ongoing in preclinical studies for validation or that are already translated in clinical research (Table 1). 

## 2. Hormone Receptors

The estrogen and progesterone receptors are overexpressed in around 75% of breast cancer either alone or together. They are considered important biomarkers to determine prognosis and predict the benefit of endocrine therapies in breast cancer patients [80,81]. Tumors with high levels of ER and PR are less aggressive and more indolent than negative tumors receptor [7]. 

Many studies report that PET imaging, using radiolabeled agents specifically targeting ER and PR, is able to detect receptor expression in tumor lesions [8]. 

Therefore, this non-invasive approach could allow the selection of eligible patients for endocrine therapies in a very early stage [82,83]. 16a-^18^F-fluoro-17b-estradiol ([^18^F]FES), an estrogen derivative PET tracer, is the most frequently used probe for ER imaging applications. Clinical studies have revealed that [^18^F]FES is useful to detect positive lesions both in primary and metastatic breast cancer and in predicting response to anti-ER treatments with a good sensitivity (69–100%) and high specificity (80–100%) [9,10,11,12]. Therefore, [^18^F]FES-PET is an appropriate imaging tool for the diagnosis, staging, and post-therapeutic evaluation of patients with ER+ breast cancer [7]. It has been reported that this tracer shows greater benefits than [^18^F]-FDG-PET to distinguish flare reaction from a real disease progression [13]. Furthermore, a preoperative phase study by Gemignani et al. demonstrated that [^18^F]FES-PET Standardized Uptake Value (SUV) correlated with ER immunohistochemistry status in patients with early breast cancer and with size of the primary lesion [11]. 

Other findings have validated the association of [^18^F]FES uptake with ER concentration and the capability of this tracer to differentiate between positive and negative ER breast carcinomas [8]. At staging, a high uptake of [^18^F]FES is predictive of a good therapeutic response and an early detection of ER levels could increase treatment success. Furthermore, it has been described that [^18^F]FES-PET may help in identifying patients with endocrine-resistant metastatic breast cancer that may paradoxically benefit from estradiol therapy [14]. Recently, Wang et al. reported that [^18^F]FES-PET/CT could be helpful to assess both the receptor status and the very early response to treatment with a novel ER antagonist GDC-0810 (Figure 1) [15]. The use of this tracer in a phase I dose escalation trial of GDC-0810 was capable to define the best dose of drug necessary to suppress ER thus reducing dose-related toxicities [15]. Importantly, several findings showed that PET imaging using ^18^F-FES might be advantageous in patients with multiple lesions that are difficult to biopsy [12,16]. A novel estrogen PET tracer, 4-fluoro-11β-methoxy-16α- ^18^F-fluoroestradiol ([^18^F]4FMFES), was evaluated by Paquette et al. first in a murine model of breast cancer and then in a clinical study [17,18]. The authors compared the diagnostic potential of [^18^F]4FMFES with [^18^F]FES in a phase II clinical trial carried out in ER+ breast cancer patients. Here, they showed that [^18^F]4FMFES detected more lesions in comparison to [^18^F]FES with less background activity thus increasing diagnostic confidence and reduce false-negative diagnoses. 

Many synthetic progestins with strong affinity for PR have been used as imaging agents. 21-[^18^F]Fluoro-16α-ethyl-19-norprogesterone ([^18^F]FENP) was the first radiolabeled progestin to be widely studied. Unfortunately, the high uptake of this tracer in breast adipose tissue caused its clinical application failure [19]. Nowadays, 21-[^18^F]-fluoro-16α,17α-[(R)-(1’-α-furylmethylidene)dioxy]-19-norpregn-4-ene-3,20-dione ([^18^F]FFNP) is the PET tracer most used for PR imaging [20,21]. Its dosimetry, safety, and correlation with PR status were evaluated in a first-in-human study performed in a small cohort of primary breast cancer patients [83]. Interestingly, a preclinical study carried out on endocrine-sensitive ERα+ MCF7 human breast xenografts, showed that the reduction of [^18^F]FFNP uptake after estrogen-deprivation therapy, compared to baseline scan, predicted a positive tumor response [22]. Conversely, Salem K et al. observed in T47D breast tumor xenografts treated for only 3 days with 17β-estradiol, an increase of [^18^F]FFNP uptake in comparison to vehicle treated animals and a corresponding enhancement of PR levels [23]. Noteworthy, an advantage of [^18^F]FFNP over ^18^F-FES is its ability to visualize lesions and measure PR status in patients treated with ER blocking agents, such as tamoxifen. Currently, in an ongoing clinical trial (NCT02455453; Clinicaltrials.gov) [^18^F]FFNP uptake is being assessed as a potential predictive biomarker of endocrine therapy response in ER+ metastatic breast cancer patients. Many studies are underway to validate the use of these tracers targeting hormone receptors alone or in combination to guide patient management towards the most efficient and least toxic therapies. An extensive and detailed review, focusing only on PET imaging of ER and PR, is beyond the scope of the present paper and we refer the reader to a recent and excellent review of this topic [8].

## 3. Receptor Tyrosine Kinases

Receptor tyrosine kinases (RTKs) are often aberrantly activated during cancer progression and are involved in mediating signaling of all cancer hallmarks [84]. These receptors include an extracellular ligand-binding domain, a single transmembrane domain, a cytoplasmic kinase domain, and a carboxyl terminal tail. Generally, their activation is induced by specific ligands through dimerization and auto-phosphorylation. In many carcinomas, the gain-of-function mutations cause the constitutive activation of RTK and an amplification of intracellular signal pathways. Different classes of RTKs are deregulated in breast carcinoma including: the epidermal growth factor receptor family (EGFR/erbB1/HER1, Neu/erbB2/HER2, erbB3/HER-3, and erbB4/HER-4); vascular endothelial growth factor receptor (VEGFR); hepatocyte growth factor receptor (HGFR/c-Met); platelet-derived growth factor receptors (PDGFRs) and insulin-like growth factor-1 receptor (IGF-1R). 

Many molecules, in particular antibodies, have been developed as imaging agents to detect HER2 expression in breast cancer lesions. Perik et al. showed that scintigraphy with ^111^Indium [^111^In]trastuzumab identified HER2-positive tumors but it was not useful in predicting trastuzumab-related cardiotoxicity in metastatic breast cancer patients [24]. Similarly, in a feasibility study on 14 patients with metastatic breast cancer, PET scan performed after administration of ^89^Zirconium [^89^Zr]trastuzumab allowed the visualization and quantification of the tracer uptake in HER2-positive lesions [25]. A very interesting study demonstrated that [^89^Zr]trastuzumab is able to detect the inter-lesion heterogeneity in advanced disease and select patients who would benefit from HER2-targeting therapy with antibody-drug conjugate trastuzumab emtansine (T-DM1) [26]. Furthermore, it was reported that this tracer could differentiate HER2-positive from HER2-negative and identify intra-patient heterogeneity in 20% of patients bearing numerous lesions [27]. Noteworthy, [^89^Zr]trastuzumab was capable of detecting HER2-positive metastases in a group of patients with HER2-negative primary breast cancer [28]. Furthermore, Bensch F et al. reported that [^89^Zr]trastuzumab PET could support clinical decision-making when standard approaches are not suitable to determinate HER2 status and repeat biopsies are not possible [29]. A representative image of [^89^Zr]trastuzumab tumor uptake in a HER2-positive breast cancer patient is reported in Figure 2. Similarly, [^64^Cu]DOTA-trastuzumab visualized HER2-positive metastatic breast cancer with high precision thus providing valuable information for selection of patients that could benefit by HER2-targeted treatments [30,31]. Recently, pertuzumab, another newer humanized HER2-targeting antibody, was labeled with ^89^Zr and used for PET imaging of HER2-positive breast cancer patients. Since pertuzumab is able to bind HER2 at different sites and more efficiently compared to trastuzumab, the employment of this tracer could improve the detection of HER2-positive lesions that are not detected using trastuzumab [32]. This first-in-human trial demonstrated that [^89^Zr]pertuzumab PET/CT may be safely performed and it is able to evaluate the HER2 status and heterogeneity of lesions and detect unsuspected HER2-positive metastatic disease thus helping direct HER2-targeted therapy to appropriate patients [32]. Furthermore, pre-clinical findings showed that this tracer monitored early response to T-DM1 therapy of HER2-positive breast cancer xenograft-bearing mice and allowed precise detection of changes in tumor volume [33]. Many studies are ongoing to develop new HER2 targeting probes including antibody fragments, affibodies and nanobodies with superior characteristics appropriate for imaging, such as rapid targeting and quick blood clearance, high solubility, high stability, higher target-to-background ratios. The first-in-human application of ^68^Gallium [^68^Ga]HER2-nanobody showed a favorable biodistribution, with the highest uptake in the kidneys, liver, and intestines and low background levels in other organs that generally harbor tumor lesions [34]. In a phase I/II study, the HER2-binding affibody ABY-025 was labeled with ^68^Ga and used to measure HER2 levels in metastatic in breast cancer. SUV correlated with biopsy HER2-scores (*r* = 0.91, *p* < 0.001) and PET correctly identified the conversion and mixed expression of HER2. These observations allowed the possibility to change the treatments in 3 of the 16 patients [35]. The tumor-targeting potential of the anti-HER2 nanobodies 5F7 and 2Rs15d labeled with ^18^F were assessed in a subcutaneous HER2-positive breast cancer murine model and in brain metastases [36,37]. Notably, it has been reported the involvement of HER3 in the development of the resistance to anti-HER2 therapies in breast cancer [85]. A specific monoclonal anti-HER3 antibody, patritumab, was labeled with ^64^Cu and its safety, dosimetry, and binding were assessed in a group of patients with advance solid tumors including breast cancer [38]. Wehrenberg-Klee et al. demonstrated in a preclinical study that SUVmean of [^64^Cu]anti-HER3-F(ab′)2 increased in MDAMB468 xenografts treated with the AKT (Protein Kinase B) inhibitor GDC-0068 when compared to untreated control. The enhancement of tracer uptake in tumor correlated with HER3 levels and resistance to therapy [39]. 

The Vascular endothelial growth factor/Vascular endothelial growth factor receptor (VEGF/VEGFR) axis takes part in the regulation of angiogenesis in breast cancer, therefore, several molecules were developed to target it and then tested as anticancer agents and imaging probes [86]. 3-Piperidinylethoxy-anilinoquinazoline (PAQ) an analog to TKI vandetanib, with 40-fold stronger inhibitory properties for the VEGFR-2, was labeled with ^11^C and used as PET probe for monitoring anticancer treatment in the MMTV-PyMT/FVB (PyMT) transgenic mouse breast cancer model [40]. Mice treated with vehicle, or the anti-VEGFA murine antibody B20-4.1.1, or paclitaxel (PTX) in combination or as single agents showed a SUVmax significantly reduced after 4 days in the B20-4.1.1/PTX combinational and B20-4.1.1 monotherapy groups (*p* < 0.0005 and *p* < 0.003, respectively). 

The Hepatocyte growth factor (HGF) receptor (c-Met) is overexpressed in basal-like phenotype of breast cancer that includes the aggressive TNBC sub-group [41,42]. PET imaging with [^18^F]AH113804 peptide, which has high affinity for human c-Met allowed the early recognition of locoregional tumor recurrence in a human basal-like murine breast cancer model [43]. The expression of the insulin-like growth factor 1 (IGF-1R) is assumed to be linked with the overall survival of breast cancer patients. Furthermore, phosphorylated IGF-1R appears to be an encouraging indicator for predicting clinical outcomes and may be an attractive target to improve antitumor treatment efficacy in patients with HER2−, ER+, and luminal B tumors. [87]. The antibody R1507 targeting IGF-1R and labeled with ^111^In for SPECT and ^89^Zr for PET imaging, respectively, clearly visualized the subcutaneous TNBC SUM149 xenografts [44]. In the same animal model [^111^In]F(ab′)2 fragments showed an improved ability to target tumors expressing IGF-1R and a higher tumor-to-blood ratio in comparison to [^111^In]R1507 [45]. Many fibrotic diseases and malignant tumors are associated with platelet-derived growth factor receptor beta (PDGFRβ) overexpression and disproportionate signaling, making this receptor attractive for molecular targeting and imaging approaches. Recently, in a subgroup of mesenchymal TNBCs with invasive and stem-like phenotype, the role played by PDGFRβ as a suitable biomarker was investigated and the aptamer Gint4.T specifically targeting this receptor was proposed as a high effective tool for the imaging and suppression of TNBC lung metastases [88,89,90].

## 4. Integrin Receptors 

The integrin families are heterodymeric cell adhesion receptors composed of non-covalently associated α and β subunits that promote cell attachment and migration on the neighboring extracellular matrix (ECM). They regulate diverse cellular functions such as migration, invasion, proliferation, and survival, which are crucial to the initiation, progression, and metastasis of solid tumors [91]. 

In several tumor types, including breast cancer, the expression of specific integrins is associated with a worse patient prognosis and survival [91]. Integrins are capable of clustering together growth factors receptors in the membrane where they physically interact with each other to coordinate different pathways [92]. Interestingly, Stewart RL et al. observed a cross-talk between integrin α6β4 and HER2 in breast cancer [93]. Recently, it was reported that integrin αvβ3 is correlated with EGFR when TNBC cells were cultured on matrigel or injected subcutaneously in nude mice to form tumors, and this interaction was hampered by treatment with a specific EGFR aptamer CL4 both in vitro and in vivo [94]. This integrin is significantly up regulated on activated-endothelial cells and not on quiescent endothelial cells [95] and plays a crucial role in promoting tumor neo-angiogenesis and metastasis in aggressive tumors [91]. Therefore, labeled integrin antagonists could provide interesting diagnostic tools to assess the effectiveness of new anti-angiogenic and anti-tumoral therapies in breast cancer. Many molecules targeting αvβ3 and harboring the same RGD (arginine-glycine-aspartic acid) sequence present in some ECM proteins, such as vitronectin and fibronectin, have been designed and developed as instruments for imaging and therapy [96,97,98,99,100]. ^18^F-Galacto-RGD was the first PET tracer applied in patients which successfully imaged αvβ3 expression in human tumors including breast carcinomas with excellent tumor-background [46]. Several imaging studies using other RGD-containing radiolabeled peptides, such as [^18^F]AH111585 and [^99m^Tc]NC100692, were carried out to visualize malignant lesions in small cohorts of patients with breast cancer [47,48]. Moreover, variations in tracer design have been introduced to further improve the performance of αvβ3 imaging, e.g., the development of multimeric-RGD peptides able to recognize multivalent binding sites in RGD-containing ECM-proteins [101]. A pEGylated-dimeric-RGD (PEG3-E[c{RGDyk}]2) (FPPRGD2) labeled with [^18^F] showed high and specific uptake in the primary tumor as well as in the metastases, with no safety issues in breast cancer patients. In addition, this probe showed higher average SUVmax when compared to monomeric RGD peptides [49]. [^99m^Tc]3PRGD2 tracer was evaluated for differential diagnosis of breast cancer and compared with traditional mammography [50]. When two methods were combined, the accuracy of diagnosis of breast cancer lesions enhanced greatly, especially for the dense mammary gland. Furthermore, [^99m^Tc]3PRGD2 imaging had good performance for diagnosing primary breast tumor compared to [^18^F]FDG PET/CT but showed lower sensitivity for identifying smaller metastatic lesions in the lymph nodes [51]. Moreover, the tracer uptake was related with integrin αvβ3 expression (*r* = 0.582; *p* = 0.001) and was greater in HER2-positive and stage III–IV patients (*p* < 0.05) [51]. A correlation between [^18^F]FDG and [^68^Ga]RGD in patients with large or advanced invasive ductal breast cancer showed that SUVavg and SUVmax for [^18^F]FDG was significantly higher in the ER negative and PR negative groups whereas these same parameters were significantly higher for [^68^Ga]RGD in the HER2-positive group [52]. Recently, a fluoride-aluminum complex, [^18^F]Al-NOTA-PRGD2 (denoted as ^18^F-Alfatide II), was developed and used in differentiating breast cancer from benign breast lesions [53]. Breast carcinoma showed relatively higher tracer uptake with a SUVmax of 3.77 ± 1.78 and SUVmean of 2.25 ± 0.98 compared to benign breast lesions. Furthermore, the authors showed that [^18^F]Alfatide II had comparable diagnostic value to [^18^F]FDG, but it was not superior in identifying breast cancer lesions. Interestingly, a heterodimeric peptide was synthesized from bombesin(7–14) (BBN) and c(RGDyK) and labeled with [^68^Ga], ([^68^Ga]BBN-RGD), to target gastrin-releasing peptide receptor (GRPR) together with integrin αvβ3 [54,55]. This tracer showed several advantages compared to the corresponding monomers because of the multivalency effects that result in an improved binding affinity and increased number of effective receptors [54,55] (Figure 3). Significant correlations were identified between SUVmean determined by [^68^Ga]BBN-RGD PET and GRPR (*r*^2^ = 0.4791, *p* < 0.0001) and integrin αvβ3 (*r^2^* = 0.3664, *p* < 0.001). The correlation was greater when the sum of integrin αvβ3 and GRPR was applied. [^68^Ga]BBN-RGD PET/CT was capable of visualizing primary breast cancer, axillary lymph nodes, and distant metastases. Recently, Li et al. provided pilot clinical evidence that tumor [^18^F]RGD uptake parameters at baseline are predictive for the therapeutic response to apatinib, an oral antiangiogenic drug, in breast cancer patients with local or metastatic lesions [56]. Patients whose lesions had a higher SUVmean responded better to treatment and showed a longer progression-free survival than those with a lower SUVmean. 

Many reports have shown that tumors originating from the breast express elevated levels of αvβ6 integrin, therefore a specific peptide that targets this receptor has been developed and labeled with ^18^F for PET imaging. In a first-in-human study, the uptake of this probe was detected in primary breast lesions and lymph node metastases in stage IV invasive breast cancer [57].

## 5. Chemokine Receptors

Chemokines belong to a family of molecules that regulate many cellular functions by binding to G-protein-coupled receptors (GPCRs) with seven transmembrane domains. Upon binding to their specific ligand, they undergo a conformational change leading to the activation of intracellular signal pathways that evoke different cellular responses. The chemokine system comprises almost 50 chemokines and approximately 20 chemokine receptors. These receptors are classified as CC, CXC, CX3C, or XC based on the subfamily of their preferred ligands [102]. Interaction between chemokines and their specific receptors are critical to different biological processes in cancer progression such as tumor cell growth, tumor angiogenesis, organ-directed metastasis [103]. Particularly, chemokines and chemokine receptors such as CXCL12/CXCR4-CXCR7, CXCL16/CXCR6, CCL25/CCR9, CCL2/CCR2, and CCL5/CCR5 are found to play a crucial part in breast cancer progression [104]. 

The involvement of CXCL12/CXCR4 axis is well known in metastatic spreading, drug resistance, and overall worse diagnosis in breast carcinoma [105,106]. Furthermore, it facilitates cancer cell survival by allowing immune evasion and creating a favorable microenvironment for cancer cells to grow within the metastatic niche [107]. Promising molecules targeting CXCR4 have been developed for imaging and therapy [108,109]. A bicyclam-based antagonist AMD3100 (Plerixafor) was the first drug designed to block CXCR4 and to be used in humans [110]. Recently, Pernas et al. reported a phase 1 activity of balixafortide (CXCR4 antagonist) in combination with eribulin in heavily pretreated, relapsed metastatic breast cancer patients [111]. The tolerability and safety of this combination (balixafortide and eribulin) was comparable to that of each individual, and these two drugs together showed promising results in HER2 negative metastatic breast cancer patients. The application of molecular imaging with probes targeting CXCR4 could help to differentiate patients, monitor anti-CXCR4 therapy, and to clarify the intricate systemic effects. To visualize CXCR4 expressing tumor lesions, CXCR4 inhibitor AMD3100 was radiolabeled with [^64^Cu] and used for PET imaging. In preclinical studies, [^64^Cu]AMD3100 was able to image breast cancer xenografts as well as lung metastases [58]. Similarly, tumor uptake of a ^64^Cu doped gold nanoclusters conjugated with AMD3100 ([^64^Cu]AuNCs-AMD3100) tracer correlated with CXCR4 expression in a 4T1 orthotopic mouse breast cancer. Furthermore, this probe was better than [^64^Cu]AMD3100 alone in detecting initial-stage lesions and pre-metastatic niches in lungs [59]. A first pilot study with [^68^Ga]Pentixafor was carried out in a limited and heterogeneous group of patients with solid tumors including three patients with breast cancer [60]. Compared with [^18^F]FDG PET, this tracer showed a lower SUVmax in all measured malignant lesions. Afterwards, these observations were confirmed on a broader range of breast carcinomas with different histological features where [^68^Ga]Pentixafor uptake in tumor lesions was considerably lower compared to [^18^F]FDG [61] (Figure 4). Noteworthy, it has been reported that CXCR4 signaling could promote ER positive breast cancer to a therapy-resistant, estrogen-independent phenotype [112]. Therefore, imaging of CXCR4 in these circumstances may benefit from longitudinal spatio-temporal data on tumor development/metastasis during estrogen deprivation therapy. Recently, a cyclam-based small molecule radioprobe [^18^F]MCFB was developed and evaluated for imaging of CXCR4 expression in breast carcinoma xenografts [62]. The atypical chemokine receptor ACKR3 (CXCR7) is undetectable in normal breast tissues whereas is overexpressed in more than 30% of breast carcinomas [113]. Furthermore, the overexpression of this receptor is associated with poor overall survival and reduced lung metastasis–free survival in patients with invasive ductal carcinomas and with reduced relapse-free survival in patients with ER positive breast cancer [114]. In vivo [^89^Zr]ACKR3-mAb PET and ex vivo biodistribution studies performed in mice harboring TNBC tumors with graded ACKR3 expression levels, showed a direct correlation between tracer uptake and ACKR3 levels [63]. The signal specificity was verified through blocking studies performed in the presence of unlabeled ACKR3-mAb, which caused a significant reduction of tracer uptake in high-ACKR3-expressing tumors.

## 6. Immune Checkpoint Receptors 

Tumors avoid the host immune system by activating biological systems, which in turn suppress antigen exposition and effector cells and create an immunosuppressive milieu in the cancer microenvironment [115]. Cancer cells activating inhibitory immune checkpoint proteins such as PD-1 (program death 1) receptor and PD-L1 (programmed cell death-ligand 1) evade and suppress immune responses [116]. PD-1 receptor is a trans-membrane receptor present on a variety of immune cells, such as monocytes, T cells, B cells, dendritic cells, and tumor-infiltrating lymphocytes (TILs). Conversely, PD-L1 is expressed by tumor and antigen presenting cells (APCs) and the binding of PD-L1 to PD-1 present on T cells leads to their dysfunction, exhaustion, and neutralization [117]. Therefore, PD-L1 overexpression protects tumor cells from T cell (CD8+) cytotoxicity [118]. In addition, B7-1 (CD80), another molecule expressed by T cells and APCs, is able to interact with PD-L1 promoting a negative regulation of T cell activation [119]. In this manner tumors create and uphold an immunosuppressive environment, which endorse cancer cell escape from immune surveillance. Immunotherapeutic approaches aim to trigger adaptive and native innate anti-tumor by reversing tumor-mediate immune suppression. 

In breast carcinoma, PD-1/PD-L1 expression has been associated with large tumor size, high grade, high proliferation, estrogen receptor-negative status, and HER2-positive status [120] and it is inversely correlated with survival [121]. In metastatic breast cancer, initial clinical trials with immune checkpoint inhibitors have shown modest but interesting responses [122]. Treatment combinations with checkpoint blockade and chemotherapy in the neo-adjuvant setting caused an enhancement of pCR rates in comparison with chemotherapy alone [122]. Currently, checkpoint inhibitors in clinical trials for aggressive metastatic breast cancer, alone or together with conventional therapies, include anti-PD-1 antibodies such as pembrolizumab and nivolumab or anti-PD-L1 antibodies such as avelumab and atezolizumab [122]. At present, routine analysis of PD-1/PD-L1 expression is performed by immunohistochemistry on bioptic samples and used to guide immunotherapy. Interestingly, current studies have spotlighted the potential of molecular imaging to visualize the levels of these receptors non-invasively in whole tumor lesions and metastases. The longitudinal monitoring of PD-L1 levels could serve as a predictive biomarker to select patients for therapy with immune checkpoint inhibitors or as a tool to monitor PD-L1 expression during conventional anticancer treatments. For the first time, Heskamp et al. showed the feasibility of imaging PD-L1 expression in tumors noninvasively using a monoclonal antibody (PD-L1.3.1) radiolabeled with ^111^In [64]. In this study, they observed using SPECT/CT that [^111^In]PD-L1.3.1 accumulated in high PD-L1-expressing breast (MDA-MB-231 and SK-Br-3) cancer xenografts while no specific uptake was observed in breast tumors with no or undetectable levels of PD-L1 (SUM149, BT474, and MCF-7). Imaging and biodistribution studies in an immune-competent transgenic mouse model of breast carcinoma, reflecting the role of PD-L1 in the immune system, were performed using a murine anti-PD-L1 antibody conjugated to [^111^In]DTPA [65]. Whole body SPECT images showed high signal intensity of [^111^In]DTPA-anti-PD-L1 in the tumors and in potentially cross-reactive organs such the thymus and the spleen [65]. Recently, an FDA (Food and Drug Administration) approved antibody atezolizumab (MPDL3280A) was used together with nab-paclitaxel for the first-line treatment of locally advanced unresectable or metastatic PD-L1-positive TNBC. This is a humanized monoclonal IgG1k antibody with high affinity for both human and mouse PD-L1. Lesniak et al. conjugated atezolizumab with DOTAGA and radiolabelled it with ^64^Cu to perform PET/CT imaging on MDA-MB-231 and SUM149 breast carcinoma xenografts and on a syngeneic orthotopic breast cancer mouse model (4T1) [66]. They showed that the uptake of [^64^Cu]atezolizumab was higher in PD-L1 overexpressing tumors (MDA-MB-231) in comparison to xenografts that express low levels of this receptor (SUM149). Comparable results were obtained when the same breast cancer animal models underwent SPECT/CT with [^111^In]atezolizumab [67]. Recently, Heskamp et al. for the first time performed imaging studies using [^111^In]-PD-L1-mAb in a humanized breast carcinoma murine harboring PD-L1 expressing immune cells. They demonstrated that despite the presence of activated PD-L1–expressing human immune cells, the visualization of PD-L1 overexpressing tumors remained feasible [68]. Furthermore, they reported that [^111^In]anti–mPD-L1 micro SPECT/CT could also sensitively monitor radiotherapy-related alteration of PD-L1 expression in tumors. Although, radiolabeled antibodies can offer data at specific doses, they have not been used to calculate the extent of PD-L1 drug engagement in patients, in part due to their prolonged circulation time. Notably, Kumar et al. reported that a high-affinity PD-L1-specific peptide (WL12) radiolabeled with ^64^Cu, was able to measure the PD-L1 engagement of each therapeutic antibody at the tumor, independently of its biophysical characteristics and in vivo kinetics [69]. These quantitative methods through mathematical modeling allow the calculation of antibody doses necessary to accomplish therapeutically effective occupancy (outlined < 90%). Noteworthy, this study demonstrates the molecular imaging approach is able to predict the necessary dose to reach effective drug efficacy assessing the binding of novel molecules to target specific receptors. 

In a first-in-human assessment of [^89^Zr]atezolizumab carried out in a group of 22 patients with bladder cancer (9 patients), NSCLC (9 patients), and TNBC (4 patients), prior to start atezolizumab therapy, tracer uptake appeared to be a good predictor of treatment response. Noteworthy, clinical responses in these patients were better correlated with pretreatment PET signal than with immunohistochemistry-or RNA-sequencing based predictive biomarkers [70] (Figure 5). 

## 7. Somatostatin Receptors, Gastrin Releasing Peptide Receptors, and Neuropeptide Y Receptors

It is well known that members of the somatostatin receptor family (SSTR) are overexpressed in neuroendocrine tumors whereas in breast carcinoma their role is not yet well clarified. Many years ago, Reubi et al. showed, through in vitro receptor autoradiography performed with a somatostatin analog labeled with [^125^I] on 342 breast cancer samples, that the levels of SSTR ranged from 21% in small tumor lesions to 46% in larger tumor lesions [123]. Furthermore, a low expression of subtype 2 SSTR was observed in normal breast tissue compared to malignant lesions [124]. A significant positive correlation between SSTR2 and ER expression was reported [125] as well as estrogen-mediated regulation of SSTR2 expression [126]. Similarly, Dalm et al. observed, in a cohort of 684 patients with LN-negative breast cancer, a strong correlation between ER, SSTR, and PR and suggested the possibility to use SSTR radiolabeled ligands to guide breast cancer therapy in estrogen-resistant patients [127]. Analogues of somatostatin such as octreotide and its derivatives, labeled with different radionuclides ^68^Ga, ^90^Yttrium [^90^Y], and ^177^Lutetium [^177^Lu] are used in the imaging and therapy of neuroendrocrine tumors [128]. Recently, several investigations have occurred using these tracers in different tumor types including breast cancer [129]. Chereau and colleagues observed that [^68^Ga]DOTATOC (DOTA-DPhe1-Tyr3—Octreotide) was able to visualize breast cancer xenografts overexpressing SSTR2 that were undetected using [^18^F]FDG [71]. An autoradiographic study on an array of human breast cancer tissues showed that the antagonist [^177^Lu]DOTABASS had greater binding affinity to SSTR2 in comparison with the established agonist radioligand [^177^Lu]DOTATATE [72]. Conversely, Dude et al. reported in a murine model of human breast cancer, overexpressing this receptor that the agonists [^68^Ga]DOTATOC (18.4 ± 2.9% ID/g) and [^68^Ga]DOTATATE (15.2 ± 2.2% ID/g, ns) had higher tumor uptake compared to [^68^Ga]NODAGA-JR11 (12.2 ± 0.8% ID/g, *p* < 0.001) [73]. 

Gastrin releasing peptide receptor (GRPR) is a G-protein coupled receptor of the bombesin family expressed in most human cancers and represents an appealing biomarker for imaging and therapy of tumors. Recently, GRPR overexpression was found in 75.8% of breast carcinomas and was highly linked with ER positivity [130]. Maina et al. introduced a novel GRPR antagonist [^68^Ga]SB3 for imaging studies and showed that it had high GRPR-affinity, good in vivo stability, and excellent targeting efficacy. In the first clinical PET/CT study carried out on a group of breast cancer patients, this tracer allowed the visualization of 50% of tumor lesions [74]. Another GRPR antagonist, RM2, was labeled with ^68^Ga and used for the PET/CT pre-treatment staging of patients with primary breast carcinoma already confirmed by biopsy [75] (Figure 6). [^68^Ga]RM2 uptakes were strongly increased in 87% of tumors in comparison to healthy breast tissue and a strong association amongst GRPR binding and the ER and PR was observed. Noteworthy, this tracer was capable of detecting internal mammary lymph nodes and contralateral axillary lymph node metastasis (verified by biopsy) and bone metastases (undetected by bone scan and CT). Recently, a retrospective study performed with [^68^Ga]RM2 and ^18^F-FDG indicated that GRPR targeting imaging could be complementary or superior to [^18^F]FDG in ER-positive tumors with a low proliferation index [131]. A prospective pilot study was performed on 35 women with suspicion of breast cancer, based on mammography or ultrasonography, using [^68^Ga]NOTA-RM26 PET/CT [76]. This tracer uptake was high in tumor lesions and correlated significantly with ER+ expression. Interesting studies evaluated a novel radiolabeled GRPR antagonist, NeoBOMB1, radiolabeled with [^67^/^68^Ga/^111^In/^177^Lu] as a theranostic tool in a prostate cancer animal model [132] and in prostate cancer patients [133]. Recently, on the basis of these observations, the same authors investigated the potential of this molecule as an agent for imaging and therapy of GRPR-positive mammary carcinoma and reported the ability of [^67^Ga]NeoBOMB1 to specifically localize in the tumors of mice bearing T-47D breast xenografts [77]. 

Notably, it has been observed that 85% of primary breast cancer and 100% of lymph node metastases express high levels of the sub-type 1 of the human neuropeptide Y receptor family [134]. This is a G-protein coupled receptor family and includes, besides Y1R, other three isoforms (Y2R, Y4R, and Y5R) [135]. In addition, Reubi et al. reported a shift from Y2R expression observed in healthy tissue to Y1R during neoplastic transformation [132]. Based on these findings, Khan and colleagues developed an analogous of the neuropeptide Y (NPY) and, after labeling it with [^99m^TC], they tested it for imaging of Y1R in four breast cancer patients showing different stages of disease [78]. They found that the tracer was able to visualize tumor and metastases whereas normal tissues showed only background radiation [78]. Interestingly, because it was observed in ER+ breast cancer cells (MCF-7) that estrogen induced up-regulation of Y1R mRNA [136], Memminger M et al. evaluated the potential of Y1R as a diagnostic target in ER+ breast cancer [137]. They investigated the effect of estradiol and tamoxifen treatment on the Y1R level in MCF-7 xenografts, through tissue autoradiography using the selective Y1R antagonist [^3^H]-UR-MK114. They found that the diagnostic value of the Y1R was compromised because anti-estrogen therapy caused receptor down-regulation. Although the majority of breast cancer tumors are ER+, in the very aggressive triple-negative forms lacking this receptor, Y1R could be an interesting imaging biomarker to investigate. Furthermore, a [^18^F]-labeled high-molecular-weight NPY glycopeptide tracer for PET imaging was developed and its ability to detect Y1R expression in vivo was tested and validated in MCF-7 tumor-bearing nude mice [79]. 

## 8. Summary

Nowadays, despite the well-known molecular heterogeneity of breast cancer and the identification of important mutations that characterize different subtypes, the only clinically relevant biomarkers and validated therapeutic targets are the hormone receptors ER, PR, and HER2. During disease progression and in response to conventional and non-conventional treatments, the receptor status can change and multiple biopsies are not always applicable. Therefore, the development of agents for nuclear imaging that allow the specific detection of receptor expression could contribute to improve patient management. In this review, we report some of the most recent studies focusing on the development and characterization of novel nuclear tracers targeting specific receptors that could be good candidates for innovative and strategic therapeutic approaches (Figure 7). 

There are many advantages that imaging techniques offer in comparison to traditional approaches that need bioptic samples. Firstly, they permit the non-invasive visualization and quantification of receptor patterning in whole-body, taking into account the inter- and intra-variability of lesions. Importantly, longitudinal studies carried out on the same patient can help clinical decision-making very early by following the evolution of receptor status. To date, [^18^F]FES and [^18^F]4FMFES are the main tracers used for ER detection in breast cancer. In addition, targeting PR using [^18^F]FFNP could be useful in-patient imaging when ER sites are saturated by tamoxifen treatment. The antibodies used in clinical trials for HER2 overexpressing tumors, trastuzumab and pertuzumab, labeled with positron radionuclides, are showing great potential as PET agents. Importantly, they are able to detect HER2-expressing metastases derived from primary negative breast cancer. To improve tissue penetration and increase signal-to-noise ratio of these tracers, smaller molecules such as antibody fragments, affibodies, and nanobodies have been developed for imaging applications and tested in pre-clinical and preliminary clinical studies. Several mechanisms are implicated in the emergence of resistance to hormone and HER2 targeting treatments, therefore some alternative targets are undergoing evaluation for the imaging and therapy of breast cancer. For example, probes targeting other RTKs besides HER2—such as HER3, VEGFR, and IGF-1R have been evaluated in breast cancer animal models. In the last decades, it has been clarified that αvβ3 integrin expression is associated with a more aggressive phenotype of breast cancer given its role in both angiogenesis and metastatic processes. In 2008, Beer et al. reported [46], for the first time, the capability of [^18^F]Galacto-RGD to visualize the expression of this receptor in breast cancer patients. Subsequently, many other tracers binding αvβ3 with improved in vivo performance have been assessed in pre-clinical studies. A recent work by Li and colleagues [56] showed how [^18^F]-ALF-NOTA-PRGD2 detecting αvβ3 could predict a better response to anti-angiogenic therapy in many solid tumors including breast cancer. The application of dual targeting probes, such as [^68^Ga]BBN-RGD, may have higher targeting efficacy aside from superior specificity than their corresponding mono-specific probes. Noteworthy, the initial results from a first-in-human study including TNBC patients showed that PET with [^89^Zr]atezolizumab was able to detect PD-L1 status and predict response to atezolizumab therapy. Furthermore, Kumar et al. showed that PET imaging with a peptide radiotracer targeting PD-L1, [^64^Cu]WL12, may predict antibody dose needed to achieve therapeutically effective occupancy [69]. Moreover, other receptors such as SSTR, GRPR, and Y1R have been described to be also overexpressed in breast cancer, and radiotracers targeting them are used ongoing in preclinical and in-first-human clinical studies.

## 9. Conclusions and Future Perspectives

Many studies discussed in this review evaluated the same molecule both as an imaging agent and as an anti-cancer drug thus performing a theranostic approach (Table 2). This is an emerging field of medicine that by combining diagnostic tools and therapeutic strategies allows simultaneously or sequentially diagnosis, treatment, and monitoring drug response [138,139]. Considering that, with the exception of hormone receptors, most of the theranostic agents are currently evaluated in pre-clinical stages or in small groups of patients, further studies are still needed to translate them for clinical use. However, there are encouraging data suggesting that nuclear imaging may be a valid instrument to select breast cancer patients that could benefit from receptor targeted therapies and to monitor very early therapeutic response. Several studies are ongoing to validate promising probes that could help clinicians choose tailored treatments for each patient thus reducing therapeutic failures.

## Figures and Tables

**Figure 1 cancers-11-01614-f001:**
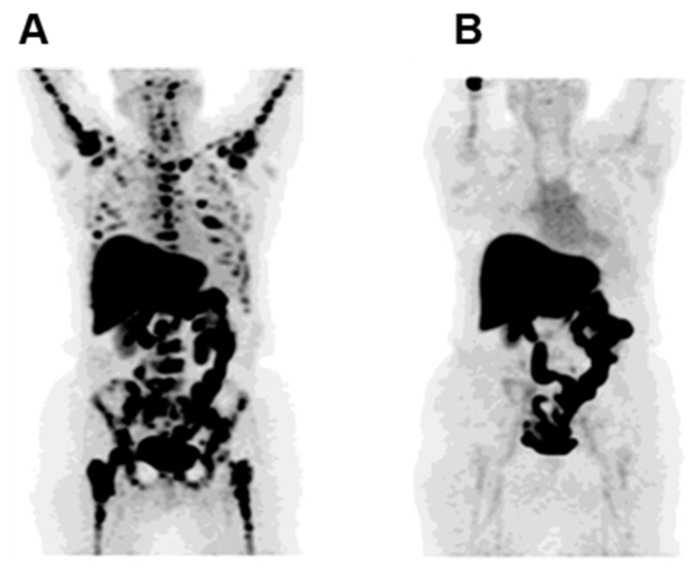
Complete suppression of 16a-^18^F-fluoro-17b-estradiol (FES)-avid lesions during GDC-0810 treatment. (**A**) Maximum intensity projection (MIP) FES- Positron Emission Tomography (PET) image before therapy showing pathologic uptake in multiple bone structures. Uptake in liver and bowel is physiologic. (**B**) MIP FES-PET image, after 2 cycles of GDC-0810 therapy, highlighted the drastic reduction of pathologic uptake (Figure adapted from Wang Y et al. (2017) [15], Permission number 4691821140364, ©AACR).

**Figure 2 cancers-11-01614-f002:**
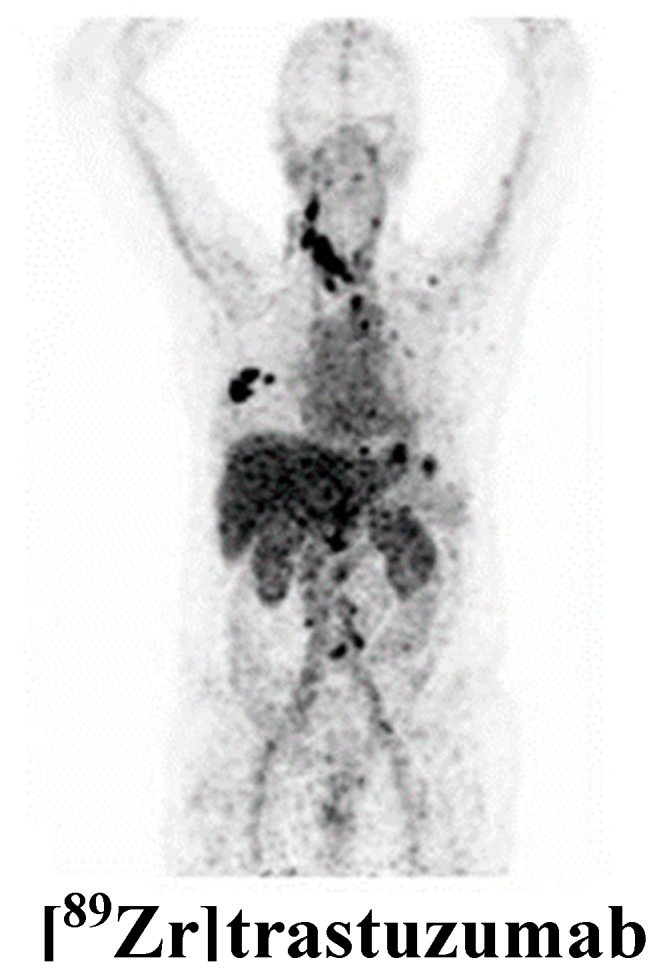
[^89^Zr]trastuzumab PET scan of a patient with HER2-positive breast cancer. (Figure adapted with permission from Bensch F et al. (2018) [29] Creative Commons Attribution 4.0 International License (http://creativecommons.org/licenses/by/4.0/).

**Figure 3 cancers-11-01614-f003:**
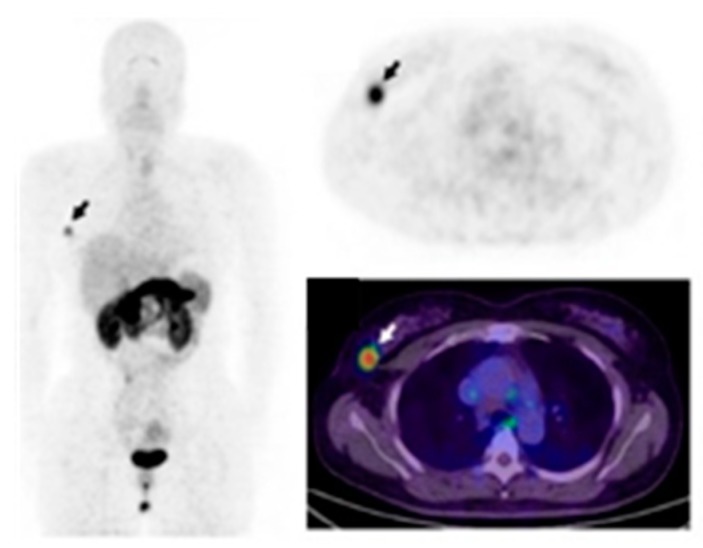
PET/CT imaging of gastrin-releasing peptide receptor and integrin αvβ3 expression using the heterodimeric tracer [^68^Ga]BBN-RGD, in a patient with ER+, PR+, and HER2+ invasive ductal carcinoma. (Figure adapted with permission from Zhang J. et al. (2018) [55] Creative Commons Attribution (CC BY-NC) license (https://creativecommons.org/licenses/by-nc/4.0/).

**Figure 4 cancers-11-01614-f004:**
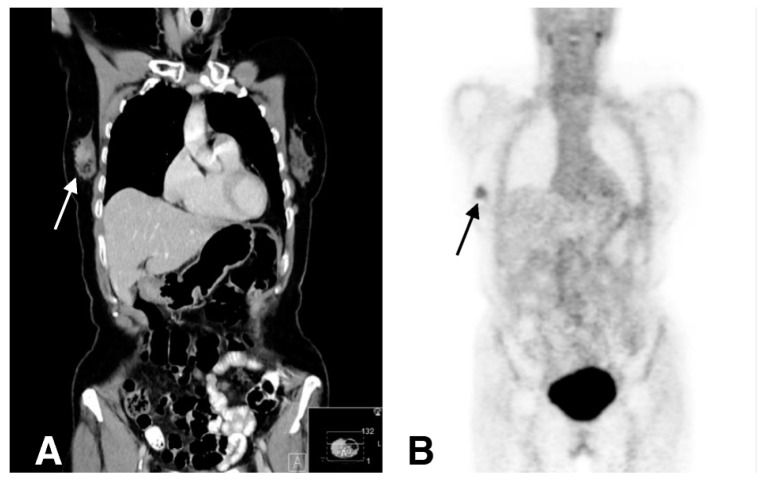
[^68^Ga]Pentixafor PET scan of a patient with invasive ductal primary breast carcinoma prior to treatment. (**A**) Coronal CT reconstruction shows contrast enhancement in a lesion in the right breast. (**B**) The tumor is visually detectable with [^68^Ga]Pentixafor. (Figure adapted with permission from Vag T. et al. (2018) [61] Creative Commons Attribution 4.0 International License (http://creativecommons.org/licenses/by/4.0/).

**Figure 5 cancers-11-01614-f005:**
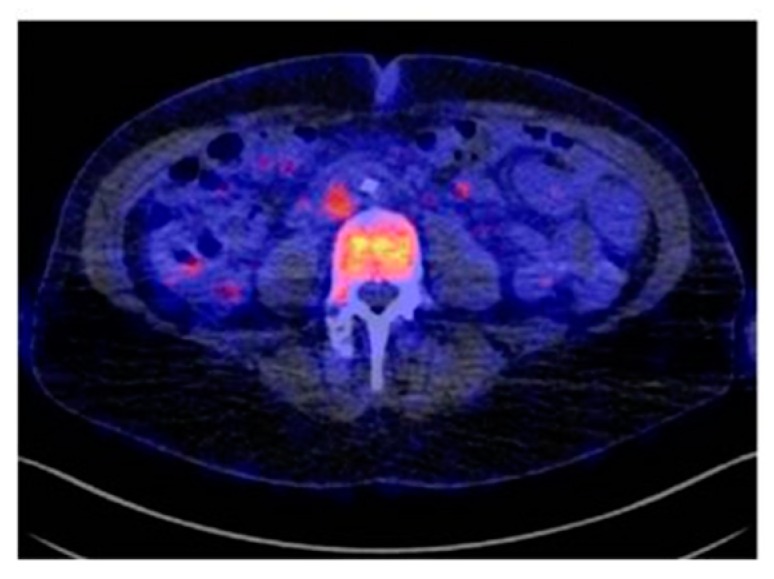
Representative PET/CT image of bone metastasis of a Triple-Negative Breast Cancer patient with [^89^Zr]atezolizumab on day 7 post injection (SUVmax 7.1). (Figure adapted from Bensch F et al. (2018) [70], Permission number 4691820293672, ©Springer Nature).

**Figure 6 cancers-11-01614-f006:**
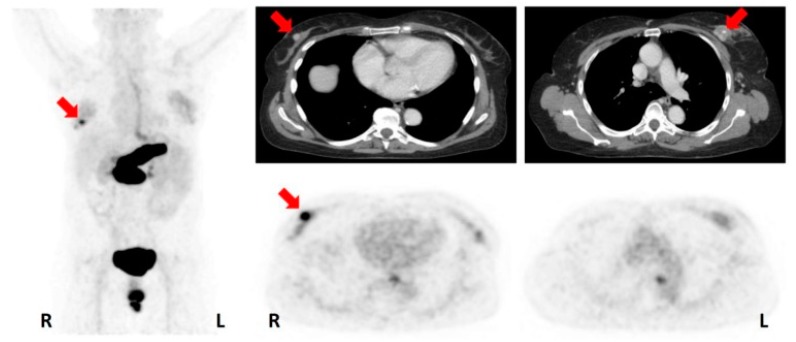
[^68^Ga]RM2-PET in a patient with bilateral invasive ductal breast carcinoma; the tumor, on the right side was ER/PR positive and showed a strongly positive uptake, while the lesion on the left side was ER/PR negative and showed a weak uptake. Pancreas, esophagus, and rectum uptake is physiological. (Unmodified from Stoykow C. et al. (2016) [75]).

**Figure 7 cancers-11-01614-f007:**
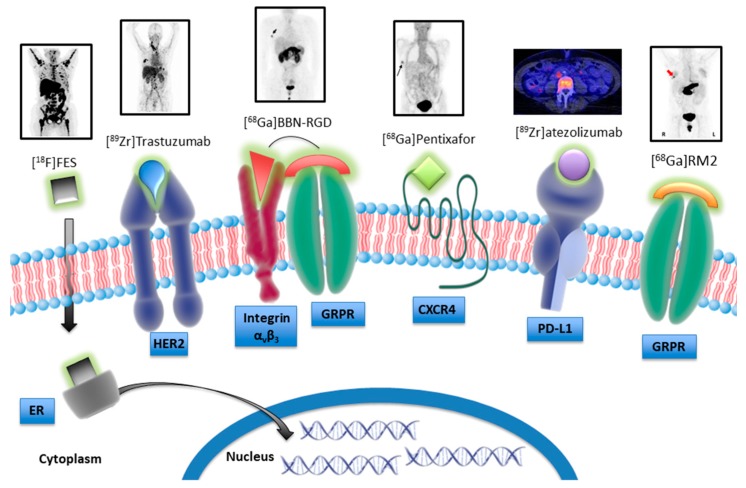
Schematic model illustrating nuclear tracers targeting specific receptors that are candidate for the theranostic approach.

**Table 1 cancers-11-01614-t001:** Summary of tracers targeting receptors for breast cancer imaging.

Receptor Type	Target	Tracer	Preclinical	Clinical	Reference
Hormone Receptors	ER	[^18^F]FES		✓	[7,8,9,10,11,12,13,14,15,16]
		[^18^F]4FMFES		✓	[17,18]
	PR	[^18^F]FENP		✓	[19,20,21]
		[^18^F]FFNP		✓	[20,21,22,23]
Tyrosine Kinase Receptors	HER2	[^111^In]trastuzumab		✓	[24]
		[^89^Zr]trastuzumab		✓	[25,26,27,28,29]
		[^64^Cu]DOTA-trastuzumab		✓	[30,31]
		[^89^Zr]Pertuzumab		✓	[32,33]
		[^68^Ga]HER2-Nanobody		✓	[34]
		[^68^Ga]ABY-025		✓	[35]
		[^18^F]-5F7	✓		[36,37]
		[^18^F]-2Rs15d	✓		[36,37]
	HER3	[^64^Cu]-patritumab		✓	[38]
		[^64^Cu]anti-HER3-F(ab′)2	✓		[39]
	VEGFR	[^11^C]-PAQ	✓		[40]
	c-Met	[^18^F]AH113804	✓		[41,42,43]
	IGFR	[^111^In]/[^89^Zr]-R1507	✓		[44]
		[^111^In]F(ab′)2	✓		[45]
Integrin Receptors	αvβ3	[^18^F]Galacto-RGD		✓	[46]
		[^18^F]AH111585		✓	[47]
		[^99m^Tc]NC100692		✓	[48]
		[^18^F]-FPPRGD2		✓	[49]
		[^99m^Tc]3PRGD2		✓	[50,51]
		[^68^Ga]RGD		✓	[52]
		[^18^F]Alfatide II		✓	[53]
		[^68^Ga]BBN-RGD		✓	[54,55]
		[^18^F]RGD		✓	[56]
	αvβ6	[^18^F]-αvβ6-BP		✓	[57]
Chemokine Receptors	CXCR4	[^64^Cu]AMD3100	✓		[58]
		[^64^Cu]AuNCs-AMD3100	✓		[59]
		[^68^Ga]Pentixafor		✓	[60,61]
		[^18^F]MCFB	✓		[62]
	CXCR7	[^89^Zr]ACKR3-mAb	✓		[63]
Immune checkpoints	PD-L1	[^111^In]PD-L1.3.1	✓		[64]
		[^111^In]- murine anti–PD-L1	✓		[65]
		[^64^Cu]Atezolizumab	✓		[66]
		[^111^In]Atezolizumab	✓		[67]
		[^111^In]-PD-L1-mAb	✓		[68]
		[^64^Cu]WL12	✓		[69]
		[^89^Zr]atezolizumab		✓	[70]
Somatostatin, Gastrin Releasing Peptide Receptors, and Neuropeptide Y receptors	SSTR	[^68^Ga]DOTATOC	✓		[71]
		[^177^Lu]DOTABASS and [^177^Lu]DOTATATE	✓		[72]
		[^68^Ga]NODAGA-JR11	✓		[73]
	GRPR	[^67^Ga]-SB3		✓	[74]
		[^68^Ga]RM2		✓	[75]
		[^68^Ga]NOTA-RM26		✓	[76]
		[^67^Ga]NeoBOMB1	✓		[77]
	NPYR	[^99m^Tc]NPYanalogue		✓	[78]
		[^18^F]NPYanalogue		✓	[79]

In this table are reported tracers currently used only in preclinical studies and tracers already translated in clinical research.

**Table 2 cancers-11-01614-t002:** Molecular Imaging for targeted therapy response assessment.

Target	Tracer	Therapy	Type of Study	Ref.
ER	[^18^F]FES	ER antagonist GDC-0810	patients with ER-positive metastatic breast cancer	[15]
HER2	[^89^Zr]trastuzumab	antibody-drug conjugate trastuzumab emtansine (T-DM1)	patients with advanced HER2-positive breast cancer	[26]
HER2	[^89^Zr]pertuzumab	antibody-drug conjugate trastuzumab emtansine (T-DM1)	mice bearing HER2+ breast cancer xenografts (BT-474)	[33]
HER3	[^64^Cu]anti-HER3-F(ab′)2	AKT inhibitor GDC-0068	mice bearing breast cancer xenografts (MDAMB468)	[39]
VEGFR	(R)-[11C]PAQ	anti-VEGFA antibody (B20-4.1.1) and/or paclitaxel (PTX)	MMTV-PyMT/FVB (PyMT) transgenic mouse breast cancer	[40]
αvβ3	[^18^F]RGD	apatinib	breast cancer patients with local or metastatic lesions	[56]
PD-L1	[^89^Zr]atezolizumab	atezolizumab	TNBC patients	[70]

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
