# Peer review of "Recent Advances in Nuclear Imaging of Receptor Expression to Guide Targeted Therapies in Breast Cancer"

_cancers, 2019, doi:10.3390/cancers11101614_

Round 1

Reviewer 1 Report

The authors have adequately addressed the points of criticism during the review process. Therefore, i recommend publication of this paper in Cancers.

Reviewer 2 Report

Your paper is well written and also responded well along the lines our reviewer commented. It is better to emphasize the utility of radiolabeled drug for the prediction of therapeutic response to save money and time since there are only 50-60% response of tamoxifen and 20% response of atezolizumab in patients with ER  and PD-L1 positives based on immunohistochemistry, respectively.

This manuscript is a resubmission of an earlier submission. The following is a list of the peer review reports and author responses from that submission.

Round 1

Reviewer 1 Report

The review paper summarizes some of the work performed both preclinically and clinically in the field of receptor expression imaging in breast cancer. 

The topic of the review is very similar to that of the 2017 paper of Dalm et al. published in ‘Int J Mol Sci.’, but given advances in the field, it could offer an updated view and potentially a broader view on receptor imaging, since in the current manuscript, additional receptors for imaging are discussed.

The current version of the paper is however not suited for publication. It lacks correct description of what is currently used as standard-of-care in breast carcinoma when it comes to receptor imaging in breast cancer. In my opinion, and looking at the EU/US guidelines of breast carcinoma management, there is none performed as standard-of-care. However, on lines 57-58, authors claim it plays “a crucial role” in the setting of metastatic breast carcinoma. Also on lines 53-54, authors claim that “non-invasive diagnostic tools to detect and visualize receptors have been greatly refined and optimized in clinical treatment strategies to improve response prediction in a more precise way. “ As these tools are in clinical trials and not used routinely worldwide in clinical treatment strategies, I believe this statement is incorrect.

Also as it comes to introduction of current treatments in breast cancer, the text should be critically reviewed with a medical oncologist treating breast carcinoma patients, to make sure that statements made by the authors are in line with current daily practice. This is currently an important lack in this manuscript.

There are also multiple parts were the text is not precise enough, where it causes confusion and questions for the reader, rather than providing answers.

For example lines 118-125 should be rewritten and made more clear for the reader: which treatment was given? What is the meaning / conclusion / potential for future patient use? Last sentence is grammatically incorrect and unclear in meaning.

Lines 458-460: it is stated that it has contributed to improve patient management. This sounds like it is already used as standard of care. Please provide more details and precision in such statements.

Overall, the text is often summing up all published data gathered by imaging of a certain receptor type in both breast carcinoma mouse models and patients, and often lacks a broader perspective of such data and its applicability in future in patients. Especially a critical view on what is really needed before it becomes standard-of-care in breast cancer patients is lacking; authors merrily state that “further studies are still needed”, without going into opportunities, hurdles and potential trial designs for imaging probes. It is clearly written by a group of preclinical imaging researchers and it misses the views of clinicians from both medical imaging and medical oncology perspective.

I would strongly recommend a very thorough rewriting of the manuscript, involving clinicians to provide the correct context and to add to the value of conclusions and future perspectives section. In addition, the figures selected for this review do not always seem to be the most appropriate ones to support the general conclusions in the review and could be improved/changed.

Major remarks:

Lines 488-489: this sentence is very confusing as it is not immediately clear that the promising data mentioned here are actually not imaging data but rather therapeutic results. Please rephrase for better understanding.

Lines 38-39: has improved outcome in metastatic BC patients: only in metastatic? I would say also and maybe even more importantly in local/locally advanced.

Lines 68-70: authors state to focus on receptor imaging using agents labeled with positron emitting radioisotopes, but subsequently in table 1, there is a mixture of PET and gamma-emitting tracers with no clear focus on the former.

Table 1 gives the impression that for those tracers already in clinical trial, there are no preclinical data available. This could be better explained in the in the text.

On line 60, ‘is’ should be changed to ‘has the potential to’, as it is currently not proven yet.

Lines 75-76 “influence around 75% of BC progression”: this is a bold statement that merits a correct reference. Or do authors merely mean that it is positive in 75% of BC?

Lines 80-81: authors claim imaging can quantify ER and PR expression. I disagree. It can assess expression as positive of negative, but as far as I am aware, there are no data supporting a real quantification. If so, please provide correct reference for this statement.

Lines 81-83: please change “allows” to “ has the potential to allow” or “could allow”, to avoid confusion that it is already current daily clinical practice to do so. Or if used in clinical routine in some centers, please indicate this.

Lines 90-93: Please change “immunohistochemistry expression” to “immunohistochemistry status”, as it is not a quantitative but a nominal measure that was assessed. Figure 2 mentioned with the reference does not provide the data claimed here by the authors.

Lines 94-99: this sentence is unclear and should be better substantiated. What is meant with “no response is detected”? In what way is the response measured in this case? If authors mean a change between baseline uptake of FES and uptaken after start of treatment, this should be made clear to the reader. The concept of evaluating receptor occupancy is not clearly explained in the manuscript. Similar comment for lines 101-106.

The usefulness of figure 2 should be critically evaluated. It merely shows the potential to detect lesions with FES, whereas the clinical potential does not lie in diagnosis but rather in response prediction. As such, a figure that demonstrates that potential would be a better choice, or should be added.

Similarly for figure 3: just showing the imaging potential of trastuzumab does not highlight its clinical potential. Showing its use in prediction of response seems much more relevant.

Lines 169-172: please specify why pertuzumab imaging adds value compared to trastuzumab imaging in patients.

Lines 404-408: is this the general conclusion for all tumor types or for breast carcinoma specifically. Please specify and add info on BC if not yet provided.

Minor comments:

HER3 is missing in figure 1 and should be added to be consistent with table 1. Figure 1 could also be omitted, as it shows the same data as provided in table 1.

Lines 165-166: metastatic as well as disseminated disease: is that not the same?

Please rephrase sentence on lines 68-69 to improve grammar and English language.

Please be consistent in methods used consensus nomenclature for isotopes and radioactive tracers, as proposed in “Consensus nomenclature rules for radiopharmaceutical chemistry — Setting the record straight”, by Coenen et al. Nuclear medicine and biology, 2017.

Line 67: primitive tumor, I guess authors mean primary tumor?

Throughout the text, “x respect to y” as in “in comparison to” is used, but this in grammatically incorrect.

Abbreviation ECM not explained.

Line 255: I believe the authors mean sensitivity in stead of sensibility?

Legend figure 4: please discuss what the imaging probe is targeting, as to allow the reader to interpret the figure without reading the entire text.

Lines 346 and 350: PD-L1 vs. PDL-1.

Spelling errors: lines 257, 380, 396, 466, 496

Spelling inconsistency: First in human vs. first-in-human; et al. vs. et al

Meaning of sentence unclear: Lines 370, 389-390, 399-402

Multiple grammatical errors: Line 145, 157; 186-187, 192-193, 209-211, 227-230, 230-231, 247-250, 418-420, 423, 430-431, 443, 445, 493-494, 496

Author Response

Naples 4th September 2019

Prof. Stefano Fanti

Metropolitan Nuclear Medicine,

University of Bologna, Bologna, Italy

Prof. Laura Evangelista

Nuclear Medicine Unit,

Veneto Institute of Oncology IOV – IRCCS, Padua, Italy

Dear Editors,

we are submitting this revised version of the Manuscript ID: cancers-554887 to Cancers - Special Issue “Role of Medical Imaging in Cancers”.

 Manuscript Title: Recent advances in nuclear imaging of receptor expression to guide targeted therapies in breast cancer

Authors: Barbara Salvatore, Maria Grazia Caprio, Billy Samuel Hill, Annachiara Sarnella, Giovanni Nicola Roviello and Antonella Zannetti.

At first, I wish to thank you and the Reviewers for the time spent on our manuscript and for the opportunity to submit a revised version of the manuscript modified in response to the comments of the Reviewers.

We are really grateful for the efforts of the Reviewers and believe that they have truly helped us to strengthen the manuscript. We are especially appreciative of your willingness to consider the paper for publication in Cancers.

Please find below the point-by-point response to Reviewer’s comments (reported in red). We highlighted the changes in red within the main body of the text.

Point-by-point response to the Reviewers

Reviewer Comments:

Reviewer 1

The review paper summarizes some of the work performed both preclinically and clinically in the field of receptor expression imaging in breast cancer.

The topic of the review is very similar to that of the 2017 paper of Dalm et al. published in ‘Int J Mol Sci.’, but given advances in the field, it could offer an updated view and potentially a broader view on receptor imaging, since in the current manuscript, additional receptors for imaging are discussed.

The current version of the paper is however not suited for publication. It lacks correct description of what is currently used as standard-of-care in breast carcinoma when it comes to receptor imaging in breast cancer. In my opinion, and looking at the EU/US guidelines of breast carcinoma management, there is none performed as standard-of-care. However, on lines 57-58, authors claim it plays “a crucial role” in the setting of metastatic breast carcinoma. Also on lines 53-54, authors claim that “non-invasive diagnostic tools to detect and visualize receptors have been greatly refined and optimized in clinical treatment strategies to improve response prediction in a more precise way. “As these tools are in clinical trials and not used routinely worldwide in clinical treatment strategies, I believe this statement is incorrect.

Also as it comes to introduction of current treatments in breast cancer, the text should be critically reviewed with a medical oncologist treating breast carcinoma patients, to make sure that statements made by the authors are in line with current daily practice. This is currently an important lack in this manuscript.

There are also multiple parts were the text is not precise enough, where it causes confusion and questions for the reader, rather than providing answers.

For example lines 118-125 should be rewritten and made more clear for the reader: which treatment was given? What is the meaning / conclusion / potential for future patient use? Last sentence is grammatically incorrect and unclear in meaning.

Lines 458-460: it is stated that it has contributed to improve patient management. This sounds like it is already used as standard of care. Please provide more details and precision in such statements.

Overall, the text is often summing up all published data gathered by imaging of a certain receptor type in both breast carcinoma mouse models and patients, and often lacks a broader perspective of such data and its applicability in future in patients. Especially a critical view on what is really needed before it becomes standard-of-care in breast cancer patients is lacking; authors merrily state that “further studies are still needed”, without going into opportunities, hurdles and potential trial designs for imaging probes. It is clearly written by a group of preclinical imaging researchers and it misses the views of clinicians from both medical imaging and medical oncology perspective.

I would strongly recommend a very thorough rewriting of the manuscript, involving clinicians to provide the correct context and to add to the value of conclusions and future perspectives section. In addition, the figures selected for this review do not always seem to be the most appropriate ones to support the general conclusions in the review and could be improved/changed.

We thank the Reviewer for his/her very careful reading of our manuscript, and for his/her important comments and suggestions. We tried to improve the manuscript considering the relevant observations made by the Reviewer. We want to clarify that main purpose of this review is to gain an overview of the relevant findings regarding the development of novel radiotracers targeting receptor status that are ongoing in preclinical and/or clinical studies as promising tools to lead treatment decisions for breast cancer management. Therefore, this manuscript was written by nuclear medicine physicians (the first authors: Barbara Salvatore and Maria Grazia Caprio), by Antonella Zannetti, PI of preclinical imaging studies in oncology (group: Billy Samuel Hill and Annachiara Sarnella) and by Giovanni Nicola Roviello with expertise in the development of novel molecules for diagnosis and therapy. Consequently, we focused on diagnostic imaging.

Major remarks:

Lines 488-489: this sentence is very confusing as it is not immediately clear that the promising data mentioned here are actually not imaging data but rather therapeutic results. Please rephrase for better understanding.

We thank the Reviewer for his/her insightful and constructive comment which allows us to clarify this point highlighting the imaging results.  In the revised manuscript we rephrased the sentence in: “Noteworthy, the initial results from a first-in-human study including TNBC patients showed that PET with [89Zr]atezolizumab was able to detect PD-L1 status and predict response to atezolizumab therapy”.

Lines 38-39: has improved outcome in metastatic BC patients: only in metastatic? I would say also and maybe even more importantly in local/locally advanced.

 We thank the reviewer for pointing this out, in light of this comment we removed metastatic from the sentence in order to include all breast cancer patients.

Lines 68-70: authors state to focus on receptor imaging using agents labeled with positron emitting radioisotopes, but subsequently in table 1, there is a mixture of PET and gamma-emitting tracers with no clear focus on the former.

According to this Reviewer’s criticism in the revised manuscript, we removed the statement that we focused only on PET tracers used for imaging of breast cancer.

Table 1 gives the impression that for those tracers already in clinical trial, there are no preclinical data available. This could be better explained in the in the text.

 According to this Reviewer’s suggestion to better clarify the data reported in Table 1, we added in the revised manuscript the following sentence at the end of the table: “In this table are reported tracers currently used only in preclinical studies and tracers already translated in clinical research”.

On line 60, ‘is’ should be changed to ‘has the potential to’, as it is currently not proven yet.

 As suggested by the reviewer this change was made in the revised manuscript.

Lines 75-76 “influence around 75% of BC progression”: this is a bold statement that merits a correct reference. Or do authors merely mean that it is positive in 75% of BC?

 As suggested by the reviewer this sentence was changed according.

Lines 80-81: authors claim imaging can quantify ER and PR expression. I disagree. It can assess expression as positive of negative, but as far as I am aware, there are no data supporting a real quantification. If so, please provide correct reference for this statement.

According to Reviewer’s criticism we removed from the sentence “…..to quantify ER and PR expression” in the revised manuscript

Lines 81-83: please change “allows” to “has the potential to allow” or “could allow”, to avoid confusion that it is already current daily clinical practice to do so. Or if used in clinical routine in some centers, please indicate this.

 As suggested by the reviewer this change was made to manuscript.

Lines 90-93: Please change “immunohistochemistry expression” to “immunohistochemistry status”, as it is not a quantitative but a nominal measure that was assessed. Figure 2 mentioned with the reference does not provide the data claimed here by the authors.

 As suggested by the reviewer this change was made in the revised manuscript. We changed the Figure 2 that now is Figure 1 as reported below.

Lines 94-99: this sentence is unclear and should be better substantiated. What is meant with “no response is detected”? In what way is the response measured in this case? If authors mean a change between baseline uptake of FES and uptaken after start of treatment, this should be made clear to the reader. The concept of evaluating receptor occupancy is not clearly explained in the manuscript. Similar comment for lines 101-106.

According to Reviewer’s criticism we removed the sentence because there are no data regarding no-responder patients to treatments and FES uptake, we reported only our hypothesis. Furthermore, we rephrased the other sentences to better clarify the concept.

The usefulness of figure 2 should be critically evaluated. It merely shows the potential to detect lesions with FES, whereas the clinical potential does not lie in diagnosis but rather in response prediction. As such, a figure that demonstrates that potential would be a better choice, or should be added.

According to this Reviewer’s suggestion, in the revised manuscript, we changed the Figure 2, that now is Figure1, and showed representative images of FES-PET scan before and after 2 cycles of GDC-0810 therapy from Clin Cancer Res. 2017, 23, 3053-3060.

Similarly, for figure 3: just showing the imaging potential of trastuzumab does not highlight its clinical potential. Showing its use in prediction of response seems much more relevant.

Unfortunately, we could not find an image with labeled-trastuzumab of a breast cancer patient after therapy. In the revised manuscript is Figure 2.

Lines 169-172: please specify why pertuzumab imaging adds value compared to trastuzumab imaging in patients.

We thank the reviewer for pointing this out, in light of this comment, we added the following sentence “Given that pertuzumab is able to bind HER2 at different sites and more efficiently compared to trastuzumab, the employment of this tracer could improve the detection of HER2+-lesions no detected using trastuzumab”.

Lines 404-408: is this the general conclusion for all tumor types or for breast carcinoma specifically. Please specify and add info on BC if not yet provided.

According to this Reviewer’s suggestion, we specified that the study was performed in 22 patients with bladder cancer (9 patients), NSCLC (9 patients) and TNBC (4 patients) and the conclusion regarding all tumor types.

Minor comments:

HER3 is missing in figure 1 and should be added to be consistent with table 1. Figure 1 could also be omitted, as it shows the same data as provided in table 1.

According to this Reviewer’s suggestion, in the revised manuscript, we removed Figure 1.

Lines 165-166: metastatic as well as disseminated disease: is that not the same?

According to Reviewer’s criticism, we removed, in the revised manuscript, “as well as disseminated disease”.

Please rephrase sentence on lines 68-69 to improve grammar and English language.

 As suggested by the reviewer the sentence was rephrased as follows: “Functional nuclear imaging using single photon emission computed tomography (SPECT) and positron emission tomography (PET) allows the visualization of molecular changes in receptor status in primary tumor, metastases and throughout treatments”.

Please be consistent in methods used consensus nomenclature for isotopes and radioactive tracers, as proposed in “Consensus nomenclature rules for radiopharmaceutical chemistry — Setting the record straight”, by Coenen et al. Nuclear medicine and biology, 2017.

As suggested by the reviewer all radiopharmaceutical nomenclature was changed according to the work by Coenen et al.in the revised manuscript.

Line 67: primitive tumor, I guess authors mean primary tumor?

We thank the reviewer for highlighting this issue and we changed accordingly in the revised manuscript.

Throughout the text, “x respect to y” as in “in comparison to” is used, but this in grammatically incorrect.

As suggested by the reviewer a careful revision of the manuscript was performed to remove these grammatical mistakes

Abbreviation ECM not explained.

The abbreviation ECM (extracellular matrix) has been explained in the revised manuscript

Line 255: I believe the authors mean sensitivity instead of sensibility?

We thank the reviewer for highlighting this issue and we changed accordingly

Legend figure 4: please discuss what the imaging probe is targeting, as to allow the reader to interpret the figure without reading the entire text.

We perfectly agree with the Reviewer and following his/her suggestion we added details to legend of figure 4 (now figure 3) thus offering to the readers a clearer outlook.

Lines 346 and 350: PD-L1 vs. PDL-1.

We thank the reviewer for highlighting this issue and we changed accordingly

Spelling errors: lines 257, 380, 396, 466, 496

We thank the reviewer for highlighting this issue and we changed accordingly

Spelling inconsistency: First in human vs. first-in-human; et al. vs. et al

We thank the reviewer for highlighting this issue and we changed accordingly

Meaning of sentence unclear: Lines 370, 389-390, 399-402

According to this Reviewer’s suggestion, we have tried to make the meaning of the sentences clearer

Multiple grammatical errors: Line 145, 157; 186-187, 192-193, 209-211, 227-230, 230-231, 247-250, 418-420, 423, 430-431, 443, 445, 493-494, 496

As suggested by the reviewer a careful revision of the manuscript was performed to remove these grammatical errors

Ultimately, we are really grateful for the efforts of the Editors and Reviewers and believe that they have truly helped us to strengthen the manuscript. We are especially appreciative of your willingness to consider the paper for publication in Cancers.

With my best personal regards

Sincerely

Antonella Zannetti, PhD

IBB - CNR Institute of Biostructures and Bioimages

National Research Council

Via T. De Amicis, 95

80145 Naples-Italy

Phone +39.081.2203431

Mobile 366.61.15.319

e-mail: antonella.zannetti@cnr.it

antonella.zannetti@ibb.cnr.it

Reviewer 2 Report

The subject of the review is targeted nuclear imaging in breast cancer. The topic is relevant but unfortunately the paper is not well written. There are a lot of grammar mistakes, and there is no clear structure. Paragraphs are not logical. Often new paragraphs are not started when this is needed, which makes the text very hard to read. 

The title of the paper suggests that "recent advances" will be discussed, but looking at the selection of papers it is unclear what the selection procedure was for the papers. The papers range from papers from the '80's - recent papers. The authors literally write "some of the nuclear tracers....". On the basis of what where "some" papers selected? This needs more explanation. This of course also translates to the body of the paper. Additionally there is no chronological order for the literature discussed and no explanation for comparisons made by the authors between radiotracers (e.g. 18F-FES vs 18F-4FMFES). I don't believe that all relevant papers are discussed.

The authors also mentioned that the focus is on PET tracers, while a lot of SPECT imaging is discussed in the text.

The introduction sections stops abruptly and needs more information on the goal of the review or at least an indication of what the review will be about. 

Not all recent radiotracers that are relevant are discussed. The section on SSTR and GRPR is much shorter and with much less detail than the other parts. Is this because not a lot of information is available of these radiotracers? I suggest to give a similar structure to each of the sections and give similar levels of detail.

In table 1 DOTA-BASS and DOTA-TATE are mentioned in one row separated by "/" which indicates that this is the same molecule, which is not the case. 

Figure 1 has no additional information and can be removed.

In the summary the text "in this review ... ... therapeutic approaches" should be moved to the introduction or to the beginning of the summary section. The concluding remarks are similar to what is also mentioned in the abstract. The summary also misses a part on SSTR and GRPR.

Other suggestions:

therapy options are discussed for ER+ tumors, therapy with anti-PDL1 etc, but theranostic options are not discussed. also consider that there are different resistance mechanisms e.g. for ER-directed treatment. In not all cases this is observerd by down regulation of the ER.

Author Response

Naples 4th September 2019

Prof. Stefano Fanti

Metropolitan Nuclear Medicine,

University of Bologna, Bologna, Italy

Prof. Laura Evangelista

Nuclear Medicine Unit,

Veneto Institute of Oncology IOV – IRCCS, Padua, Italy

Dear Editors,

we are submitting this revised version of the Manuscript ID: cancers-554887 to Cancers - Special Issue “Role of Medical Imaging in Cancers”.

 Manuscript Title: Recent advances in nuclear imaging of receptor expression to guide targeted therapies in breast cancer

Authors: Barbara Salvatore, Maria Grazia Caprio, Billy Samuel Hill, Annachiara Sarnella, Giovanni Nicola Roviello and Antonella Zannetti.

At first, I wish to thank you and the Reviewers for the time spent on our manuscript and for the opportunity to submit a revised version of the manuscript modified in response to the comments of the Reviewers.

We are really grateful for the efforts of the Reviewers and believe that they have truly helped us to strengthen the manuscript. We are especially appreciative of your willingness to consider the paper for publication in Cancers.

Please find below the point-by-point response to Reviewer’s comments (reported in red). We highlighted the changes in red within the main body of the text.

Point-by-point response to the Reviewers

Reviewer 2

The subject of the review is targeted nuclear imaging in breast cancer. The topic is relevant but unfortunately the paper is not well written. There are a lot of grammar mistakes, and there is no clear structure. Paragraphs are not logical. Often new paragraphs are not started when this is needed, which makes the text very hard to read.

We thank the Reviewer for his/her very careful reading of our manuscript, and for his/her important comments and suggestions

The title of the paper suggests that "recent advances" will be discussed, but looking at the selection of papers it is unclear what the selection procedure was for the papers. The papers range from papers from the '80's - recent papers. The authors literally write "some of the nuclear tracers....". On the basis of what where "some" papers selected? This needs more explanation. This of course also translates to the body of the paper. Additionally, there is no chronological order for the literature discussed and no explanation for comparisons made by the authors between radiotracers (e.g. 18F-FES vs 18F-4FMFES). I don't believe that all relevant papers are discussed.

We tried to improve the manuscript considering the relevant observations made by the Reviewer.

The authors also mentioned that the focus is on PET tracers, while a lot of SPECT imaging is discussed in the text.

According to this Reviewer’s criticism in the revised manuscript, we removed the statement that we focused only on PET tracers.

The introduction sections stop abruptly and needs more information on the goal of the review or at least an indication of what the review will be about.

According to this Reviewer’s suggestion, we tried to improve the manuscript by describing in more detail the aim of the review.

Not all recent radiotracers that are relevant are discussed. The section on SSTR and GRPR is much shorter and with much less detail than the other parts. Is this because not a lot of information is available of these radiotracers? I suggest to give a similar structure to each of the sections and give similar levels of detail.

As suggested by Reviewer 3, we added to this section the development of novel tracers targeting neuropeptide Y receptor 1 (Y1R) for breast cancer imaging. Unfortunately, to date there are few reports regarding the detection of SSTR, GRPR and Y1R by nuclear imaging in this neoplasia.

In table 1 DOTA-BASS and DOTA-TATE are mentioned in one row separated by "/" which indicates that this is the same molecule, which is not the case.

According to this Reviewer’s suggestion, in the revised manuscript, we removed the confusing use of “/”.  The “/” did not indicate that DOTA-BASS and DOTA-TATE are the same molecule but that both tracers were reported in the same study.

Figure 1 has no additional information and can be removed.

According to this Reviewer’s suggestion, in the revised manuscript, we removed Figure 1.

In the summary the text "in this review ... ... therapeutic approaches" should be moved to the introduction or to the beginning of the summary section. The concluding remarks are similar to what is also mentioned in the abstract. The summary also misses a part on SSTR and GRPR.

We thank the reviewer for highlighting this issue and we changed accordingly in the revised manuscript.

Other suggestions:

therapy options are discussed for ER+ tumors, therapy with anti-PDL1 etc, but theranostic options are not discussed. also consider that there are different resistance mechanisms e.g. for ER-directed treatment. In not all cases this is observed by down regulation of the ER.

We agree with Reviewer that theranostic options and other resistance mechanisms are very important issues but the focus of this review was to report the main researches in the field of nuclear imaging to assess of receptor status in breast cancer.

Ultimately, we are really grateful for the efforts of the Editors and Reviewers and believe that they have truly helped us to strengthen the manuscript. We are especially appreciative of your willingness to consider the paper for publication in Cancers.

With my best personal regards

Sincerely

Antonella Zannetti, PhD

IBB - CNR Institute of Biostructures and Bioimages

National Research Council

Via T. De Amicis, 95

80145 Naples-Italy

Phone +39.081.2203431

Mobile 366.61.15.319

e-mail: antonella.zannetti@cnr.it

antonella.zannetti@ibb.cnr.it

Reviewer 3 Report

The review by Salvatore and co-workers is concerned about "Recent advances in nuclear imaging of receptor expression in breast cancer".  This review article is well written and well organized and it provides a quick and easy overview on some interesting approaches in the field of PET imaging of breats cancer.  The authors address highly important molecular targets , such as integrins and chemokines, by discussing preclinical and clinical studies that were reported recently.  However, the field of neuropeptide receptor expression (neuropeptide-Y Y1R expression) and development of radioligands targeting these receptors seem to be neglected in this article.

I would recommend this review for publication in Cancers, subject to minor revsion:
1) The authors should include and discuss the results of  Memminger et al. in PlosOne about "The Neuropeptide Y Y1 Receptor: A Diagnostic Marker? Expression in MCF-7 Breast Cancer Cells Is Down-Regulated by Antiestrogens In Vitro and in xenografts":
 https://doi.org/10.1371/journal.pone.0051032

2) In this respect, there is important work by the Beck-Sickinger group on Tc99m-labeled peptide (also clinical studies) and F18-labeled peptides (preclinical) to target Y1R expression in breast cancer. I feel that this work should be included in the review paper, to give some information on important and intensively studied alternative GPCR targeting , which is not only SSTR and GRPR targeting,   for example: 
Hofmann et al.  Mol. Pharmaceutics 2015; 12, 4, 1121-1130.

Author Response

Naples 4th September 2019

Prof. Stefano Fanti

Metropolitan Nuclear Medicine,

University of Bologna, Bologna, Italy

Prof. Laura Evangelista

Nuclear Medicine Unit,

Veneto Institute of Oncology IOV – IRCCS, Padua, Italy

Dear Editors,

we are submitting this revised version of the Manuscript ID: cancers-554887 to Cancers - Special Issue “Role of Medical Imaging in Cancers”.

 Manuscript Title: Recent advances in nuclear imaging of receptor expression to guide targeted therapies in breast cancer

Authors: Barbara Salvatore, Maria Grazia Caprio, Billy Samuel Hill, Annachiara Sarnella, Giovanni Nicola Roviello and Antonella Zannetti.

At first, I wish to thank you and the Reviewers for the time spent on our manuscript and for the opportunity to submit a revised version of the manuscript modified in response to the comments of the Reviewers.

We are really grateful for the efforts of the Reviewers and believe that they have truly helped us to strengthen the manuscript. We are especially appreciative of your willingness to consider the paper for publication in Cancers.

Please find below the point-by-point response to Reviewer’s comments (reported in red). We highlighted the changes in red within the main body of the text.

Point-by-point response to the Reviewers

Reviewer 3

The review by Salvatore and co-workers is concerned about "Recent advances in nuclear imaging of receptor expression in breast cancer".  This review article is well written and well organized and it provides a quick and easy overview on some interesting approaches in the field of PET imaging of breast cancer.  The authors address highly important molecular targets, such as integrins and chemokines, by discussing preclinical and clinical studies that were reported recently.  However, the field of neuropeptide receptor expression (neuropeptide-Y Y1R expression) and development of radioligands targeting these receptors seem to be neglected in this article.

We thank the Reviewer for giving us the opportunity to improve our work and we hope that he/she may appreciate the newly revised manuscript.

I would recommend this review for publication in Cancers, subject to minor revisions:

1) The authors should include and discuss the results of  Memminger et al. in PlosOne about "The Neuropeptide Y Y1 Receptor: A Diagnostic Marker? Expression in MCF-7 Breast Cancer Cells Is Down-Regulated by Antiestrogens In Vitro and in xenografts":
https://doi.org/10.1371/journal.pone.0051032

2) In this respect, there is important work by the Beck-Sickinger group on Tc99m-labeled peptide (also clinical studies) and F18-labeled peptides (preclinical) to target Y1R expression in breast cancer. I feel that this work should be included in the review paper, to give some information on important and intensively studied alternative GPCR targeting, which is not only SSTR and GRPR targeting, for example: 
Hofmann et al.  Mol. Pharmaceutics 2015; 12, 4, 1121-1130

We thank the reviewer for highlighting this issue and we discussed the role of Y1R as possible imaging biomarker in breast cancer and we added the suggested papers in the revised manuscript. 

Ultimately, we are really grateful for the efforts of the Editors and Reviewers and believe that they have truly helped us to strengthen the manuscript. We are especially appreciative of your willingness to consider the paper for publication in Cancers.

With my best personal regards

Sincerely

Antonella Zannetti, PhD

IBB - CNR Institute of Biostructures and Bioimages

National Research Council

Via T. De Amicis, 95

80145 Naples-Italy

Phone +39.081.2203431

Mobile 366.61.15.319

e-mail: antonella.zannetti@cnr.it

antonella.zannetti@ibb.cnr.it

Round 2

Reviewer 1 Report

Although the authors have added some paragraphs in an effort to render the text more clear, I do not consider these changes enough for publication.

The text is in many instances still summing up findings in articles, rather than providing a clear and systematic overview of the recent literature, with prospective views on clinical use. I would strongly suggest to compare how the research in ER/PR receptors is written here, to how it has been structured in the review by Linden et al.,PET Clin. 2018 Jul;13(3):415-422. In the latter, findings are clearly much better structured and categorized, making it easy for the reader to follow the rationale of ongoing research, and making it a delight to read the text. This can not be said for the current manuscript under review. (Note: I am not an author on that review of 2018, and do not even know these authors, so this is not a subjective finding).

I would therefore recommend to reject the manuscript for publication.

May I also suggest to the authors to mark ALL changes in the text in red, even if it are only small changes, as this makes the review process much more comfortable.

Author Response

Reviewer 1

Although the authors have added some paragraphs in an effort to render the text more clear, I do not consider these changes enough for publication.

The text is in many instances still summing up findings in articles, rather than providing a clear and systematic overview of the recent literature, with prospective views on clinical use. I would strongly suggest to compare how the research in ER/PR receptors is written here, to how it has been structured in the review by Linden et al.,PET Clin. 2018 Jul;13(3):415-422. In the latter, findings are clearly much better structured and categorized, making it easy for the reader to follow the rationale of ongoing research, and making it a delight to read the text. This can not be said for the current manuscript under review. (Note: I am not an author on that review of 2018, and do not even know these authors, so this is not a subjective finding).

I would therefore recommend to reject the manuscript for publication.

May I also suggest to the authors to mark ALL changes in the text in red, even if it are only small changes, as this makes the review process much more comfortable.

I am sorry that the manuscript is not to the liking of the reviewer. However, we agree with him/her that the paper by Linden et al. is very well structured, but differently from our review this work focused only on “Clinical Potential of Estrogen and Progesterone Receptor Imaging”. According to Review’s criticism in our revised manuscript we added the following sentence “an extensive and detailed review, focusing only on PET imaging of ER and PR, is beyond the scope of the present paper and we refer the reader to a recent and excellent review of this topic [Linden et al.,PET Clin. 2018 Jul;13(3):415-422].”

We report in the revised manuscript new changes in blue.

Reviewer 2 Report

Dear authors, 

The revised version of the manuscript has drastically improved.

Some minor comments remain:

Read the manuscript carefully and correct grammar where needed. Neobomb is missing as a GRPR radiopharmaceutical for breast cancer targeting (see paper Kaloudi et al, Molecules, 2017). From the title I conclude that the goal is to discuss recent advances in target imaging to guide therapy. Please consider a paragraph with more explanation on molecular imaging, theranostics and so on. This can be described in a few sentences. The sentence "Moreover ... ... drug efficacy", would be more in place with additional information. The conclusion of the paper is a sum up of the discussed targets and respective radiopharmaceuticals. How does the reader learn what target is suitable when? Perhaps the authors can say something about which target is being studied for which subtype, disease stage etc.

Author Response

Reviewer 2

The revised version of the manuscript has drastically improved.

We thank the Reviewer for giving us the opportunity to improve our work and we hope that he/she may appreciate the newly revised manuscript.

Some minor comments remain:

Read the manuscript carefully and correct grammar where needed. Neobomb is missing as a GRPR radiopharmaceutical for breast cancer targeting (see paper Kaloudi et al, Molecules, 2017).

According to this Reviewer’s suggestion, in the revised manuscript, we reported the studies regarding NeoBOMB1 as tool for GRPR imaging.

From the title I conclude that the goal is to discuss recent advances in target imaging to guide therapy. Please consider a paragraph with more explanation on molecular imaging, theranostics and so on. This can be described in a few sentences.

According to this Reviewer’s suggestion, we discussed the importance of molecular imaging and theranostic approach in a newly added conclusion section.

The sentence "Moreover ... ... drug efficacy", would be more in place with additional information. As suggested by the Reviewer, we replaced the sentence in the paragraph “Immune Checkpoint Receptors” (pages 11 and 12; lines 430-432 of revised manuscript) where there is more additional information about the issue.

The conclusion of the paper is a sum up of the discussed targets and respective radiopharmaceuticals. How does the reader learn what target is suitable when? Perhaps the authors can say something about which target is being studied for which subtype, disease stage etc.

To date, even if the data on new theranostic molecules are encouraging, they are currently evaluated in pre-clinical stages or in small groups of patients. Therefore, further studies are needed to evaluate which target is suitable to study for which subtype, disease stage etc.